# Asymptomatic Retroperitoneal Lipoma with Extension to the Right Anteromedial Thigh

**DOI:** 10.3390/reports8030181

**Published:** 2025-09-17

**Authors:** Catalin Balta, Marian Botoncea, Lucian Toma, Rares Voda, Anastasia Balta, Cosmin Nicolescu

**Affiliations:** 1County Emergency Clinical Hospital Targu Mures, 540136 Targu Mures, Romania; catalinbalta@gmail.com (C.B.); tomalucian2002@yahoo.com (L.T.); rares.voda@umfst.ro (R.V.); nasteatrifautan@gmail.com (A.B.); cosmin.nicolescu@umfst.ro (C.N.); 2Faculty of Medicine, George Emil Palade University of Medicine, Pharmacy, Science, and Technology, 540253 Targu Mures, Romania

**Keywords:** retroperitoneal lipoma, femoral nerve compression, dual surgical approach

## Abstract

**Background and Clinical Significance:** Retroperitoneal tumors are a rare and diverse group of neoplasms, accounting for less than 1% of adult solid tumors. Retroperitoneal lipomas are particularly uncommon, with fewer than 20 cases described in the literature. Their asymptomatic growth and lack of clear anatomical boundaries can result in delayed diagnosis and substantial tumor size at clinical presentation. This case highlights a rare retroperitoneal lipoma with atypical extension into the right thigh through the muscular lacuna, mimicking a femoral hernia and compressing the femoral nerve—a presentation scarcely reported and clinically significant due to its surgical complexity and risk of recurrence. **Case Presentation:** We report the case of a 65-year-old woman from an urban setting who presented with progressive right thigh discomfort and lower limb pain during ambulation. The mass had been initially identified two years prior as a small, asymptomatic right inguinal formation during imaging to exclude an inguinal hernia. Computed tomography (CT) and magnetic resonance imaging (MRI) confirmed a large retroperitoneal lipomatous tumor extending to the anteromedial right thigh. Surgical excision was performed through a dual approach: midline laparotomy and thigh incision. A 30 × 30 cm encapsulated lipoma was removed without injuring adjacent nerves or vessels. Histopathological evaluation confirmed a mature lipoma without atypia but with a lipogranulomatous reaction. The patient’s postoperative course was favorable, with minimal residual paresthesia and complete wound healing. **Conclusions:** Although benign, retroperitoneal lipomas can mimic other pathologies and present surgical challenges when they extend beyond their typical boundaries. Early recognition and coordinated surgical management are crucial for optimal outcomes and prevention of recurrence.

## 1. Introduction and Clinical Significance

Retroperitoneal tumors are a rare and complex clinical entity, accounting for less than 1% of all solid neoplasms in adults [1]. They develop in the retroperitoneal space, a large anatomical area extending from the diaphragm to the pelvis and containing vital structures such as the kidneys, adrenal glands, pancreas, aorta, inferior vena cava, and multiple nerve and lymphatic plexuses [2]. The lack of rigid anatomical boundaries in this region allows tumors to reach considerable sizes before becoming symptomatic, often resulting in late diagnosis and substantial dimensions at the time of clinical presentation [3].

Histologically, retroperitoneal tumors are highly heterogeneous, encompassing both benign and malignant forms. Soft tissue sarcomas are the most common malignant retroperitoneal tumors, with liposarcomas and leiomyosarcomas being the most prevalent [4]. Retroperitoneal lipomas are extremely rare, with fewer than 20 cases described in the literature [5,6,7,8,9,10,11,12,13].

## 2. Case Presentation

We present the case of a 65-year-old woman from an urban area with a retroperitoneal lipoma located in the right iliac fossa that herniated through the muscular lacuna into the right Scarpa’s triangle, compressing the femoral nerve.

The patient reported that the condition was discovered incidentally two years ago, when a small tumor was detected in the right inguinal area. She consulted a surgeon and underwent a CT scan to rule out an inguinal hernia. Following the examination, a lipomatous tumor of the right thigh was diagnosed, the surgeon proposed surgery but patient refused. The disease progressed, and the mass increased in size, causing local discomfort and pain in the right lower limb during walking, which eventually prompted her to seek surgical intervention.

Family history includes a mother with type II diabetes mellitus and a father with chronic obliterative peripheral artery disease.

Personal, physiological, and pathological history includes allergy to oral antidiabetics, insulin-treated type II diabetes, grade I essential hypertension, grade I mitral regurgitation, chronic ischemic heart disease, diabetic nephropathy, diabetic neuropathy, incipient cataract, and horseshoe kidney. No additional lipomatous lesions or clinical features suggestive of systemic lipomatosis were identified in the patient. She does not smoke, consumes alcohol occasionally, and drinks one coffee per day. Her chronic medication includes clopidogrel 75 mg (0-1-0) and antihypertensive treatment.

CT (Figure 1, Figure 2, Figure 3 and Figure 4) results included a retroperitoneal tumor extending toward the thigh, requiring MRI (Figure 5, Figure 6, Figure 7 and Figure 8) for complete diagnostic evaluation.

Intraoperative frozen-section examination was not performed, as the preoperative imaging demonstrated a homogeneous fatty density on CT and MRI without suspicious characteristics, while the macroscopic evaluation revealed a well-encapsulated lesion devoid of intraoperative features suggestive of sarcomatous transformation. Based on these considerations, the surgical team elected to proceed with complete excision and capsule preservation, with the final histopathological assessment reserved for confirmation of the benign diagnosis and exclusion of liposarcoma.

A midline laparotomy was performed. Upon inspection, a retroperitoneal mass of approximately 30 × 30 cm was identified in the right iliac fossa, exerting a mass effect and displacing the right colon and intestines medially. A peritoneal incision was made lateral to the right colon in the avascular Bogros’ space on the posterior leaf, and the tumor was dissected circumferentially (located posterior to the pain triangle) without injuring nerves, vessels, or the right ureter. The tumor extended toward the anterior thigh through the muscular lacuna, necessitating a second incision on the anteromedial aspect of the right thigh in the most prominent area. The anatomical layers were dissected while preserving the right femoral nerve, which was under extrinsic compression. The extensive anatomical distribution of the lesion precluded en bloc resection, thereby necessitating a dual surgical approach to achieve complete removal. Hemostasis was ensured, and the peritoneum was closed with a continuous suture. A contact drain was placed in the right thigh. The anatomical layers were closed, followed by skin closure and sterile dressing. Monoplanar laparoraphy was performed, followed by cutaneous suture and sterile dressing. The surgical specimen (Figure 9) was sent for definitive histopathological examination. The patient’s postoperative course was favorable: intestinal transit resumed on postoperative day 2. Symptomatic treatment was administered during hospitalization. The drain was removed, and the patient was discharged 5 days after surgery.

The histopathological result (final diagnosis) was lipoma without atypia, with lipogranulomatous reaction.

Macroscopically, there were multiple tissue fragments ranging in size from 25 × 15 mm to 117 × 80 × 35 mm, with a total weight of 550 g, consisting of encapsulated adipose tissue with a smooth, yellow, elastic surface. The cut surface appeared homogeneous.

Microscopically, the tumor was composed of lobules of mature adipose cells without atypia, separated by thin fibrovascular septa. Foci of necrotic adipocytes surrounded by foamy macrophages were observed. The formation was surrounded by a fibrous capsule of variable thickness. At one month postoperation, the patient reported minor paresthesia in the medial thigh, with no other pathological findings. The surgical wound healed by first intention.

## 3. Discussion

Retroperitoneal tumors are a rare pathology that requires careful evaluation at diagnosis, with timely surgical intervention, ideally preceded by biopsy. Their progression is often asymptomatic, allowing them to reach a considerable size. The main reasons patients seek medical care are compression of adjacent organs or mechanoreceptors.

The symptomatology of retroperitoneal tumors is often nonspecific, typically resulting from compression or invasion of adjacent structures. Patients may present with vague abdominal pain, abdominal distension, constipation, intestinal obstruction, hydronephrosis, or thrombotic phenomena due to venous compression [14]. In many cases, the tumor mass is discovered incidentally during imaging investigations performed for unrelated conditions [15].

Retroperitoneal tumors are diagnosed and staged through advanced imaging studies. Contrast-enhanced computed tomography (CT) is the standard modality, providing essential information on size, extent, and relationship to major vascular structures [4]. Magnetic resonance imaging (MRI) is indicated when soft tissue characterization is crucial or when vascular invasion is suspected. Image-guided biopsy is sometimes necessary to establish a preoperative histologic diagnosis, particularly when the therapeutic approach includes nonsurgical neoadjuvant treatments [2].

The standard treatment for retroperitoneal tumors, particularly sarcomas, is complete surgical resection with negative margins (R0) [4]. Surgical intervention is often complex and may require en bloc resection of adjacent involved organs to achieve adequate oncologic margins. When complete resection is not possible, the procedure is more palliative. Radiotherapy, particularly when administered preoperatively, is increasingly used to reduce tumor size and improve R0 resection rates, although its efficacy remains debated [3].

Regarding systemic treatments, chemotherapy plays a limited role in retroperitoneal sarcomas and is primarily reserved for high-grade or advanced, metastatic cases. Recent studies have explored the role of targeted therapies and immunotherapy, but results have been modest, with limited clinical benefits [4].

The prognosis of retroperitoneal tumors is influenced by several factors, including histologic type, malignancy grade, tumor size, surgical resection radicality, and invasion of adjacent organs [16]. Overall five-year survival rates range from 50% to 70% depending on these factors, but local recurrence remains the main therapeutic challenge, with recurrence rates exceeding 40% in many series [4].

Retroperitoneal tumors should be managed at specialized centers by multidisciplinary teams, including surgical oncologists, medical oncologists, radiation therapists, and radiologists experienced in sarcoma care. This integrated approach allows the development of personalized treatment plans, optimizes outcomes, and improves patient survival [3].

Failure to perform surgery promptly can complicate the approach and prevent complete excision. In this case, the tumor was resected via two approaches, with capsular breach, which increases the risk of local recurrence. Therefore, imaging follow-up at 6–12 months postoperatively is necessary. Our case is quite similar to the case published by Jason R Laurens et al. [17].

A review of the literature indicates that retroperitoneal lipomas are exceedingly rare, with fewer than 20 cases reported to date, most of which presented at an advanced stage owing to vague or absent clinical manifestations [5,6,7,15,17]. Tumor dimensions vary substantially, ranging from 10 cm to over 30 cm, with surgical excision consistently identified as the treatment of choice. While the majority of retroperitoneal lipomas follow a benign course, the principal diagnostic challenge lies in distinguishing them from well-differentiated liposarcomas, given the considerable overlap in radiological characteristics. Intraoperative frozen-section analysis is frequently inconclusive, rendering definitive histopathological evaluation mandatory for diagnosis [2,3,4].

## 4. Conclusions

Lipomatous retroperitoneal tumors are often asymptomatic and can grow to a substantial size before detection. They may be discovered only when they begin compressing adjacent structures or extend into other regions, such as herniating through the muscular lacuna into the thigh—as seen in this case—where they can mimic an inguinal or femoral hernia.

From a pragmatic surgical standpoint, management of such cases should include (1) comprehensive preoperative assessment with CT and MRI to delineate the lesion; (2) selective use of biopsy, reserved for cases in which sarcoma-directed neoadjuvant therapy is under consideration, consistent with current sarcoma management protocols; (3) complete surgical excision with meticulous preservation of adjacent neurovascular structures whenever feasible; and (4) structured postoperative surveillance with cross-sectional imaging at 6–12 months to detect recurrence. This algorithm provides a rational framework for the surgical management of rare benign retroperitoneal tumors with atypical anatomical extension, as exemplified by the present case.

The take-home message is that the early identification, meticulous preoperative planning, and complete surgical excision followed by structured postoperative surveillance are essential to achieving optimal outcomes in patients with giant retroperitoneal lipomas.

## 5. Limitations

This report describes a single retrospective case, which inherently restricts the generalizability of its findings. Although case reports play an important role in highlighting rare clinical entities and surgical challenges, they are limited by the absence of control groups and the inability to define causality or establish standardized management protocols. As underscored in the surgical literature, such reports should therefore be interpreted with caution, serving primarily to raise clinical awareness and generate hypotheses rather than to provide definitive therapeutic guidance [18].

## Figures and Tables

**Figure 1 reports-08-00181-f001:**
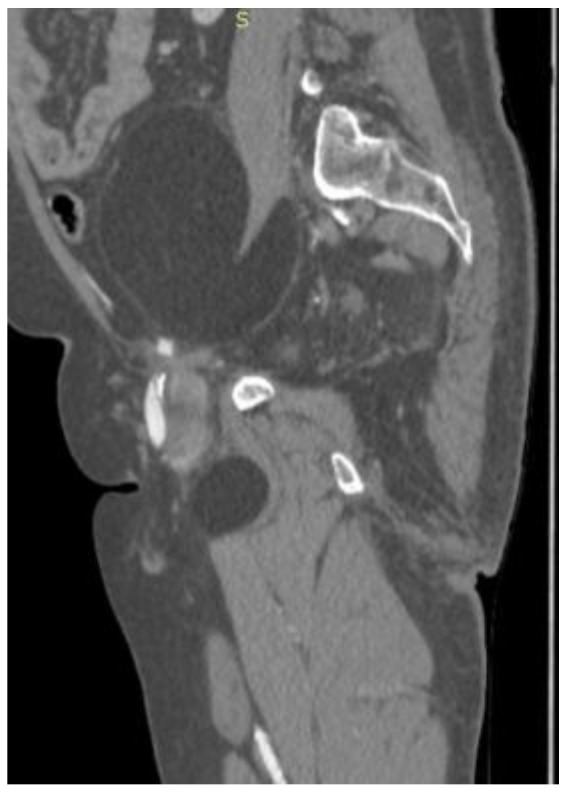
Sagittal contrast-enhanced CT image showing a large, well-encapsulated retroperitoneal mass of homogeneous fatty attenuation, extending from the abdominal cavity into the proximal thigh, displacing adjacent structures without signs of local invasion.

**Figure 2 reports-08-00181-f002:**
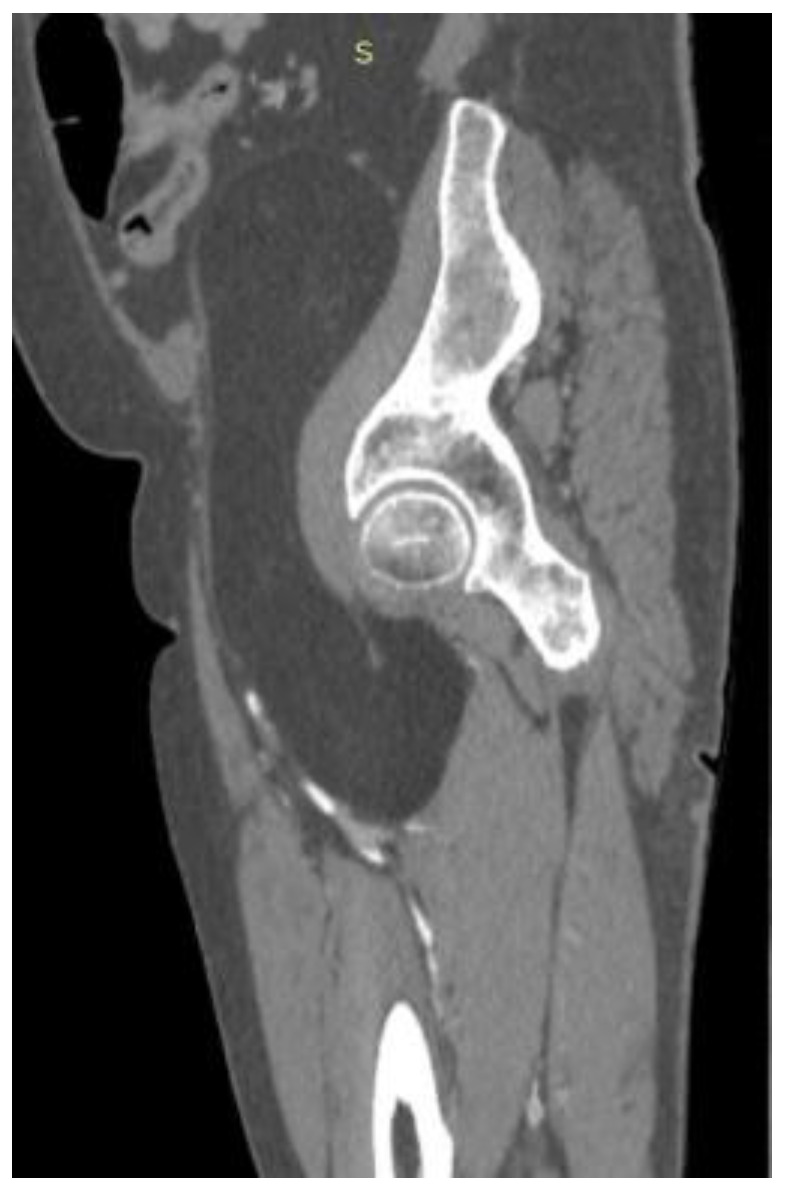
Sagittal CT reconstruction demonstrating a well-defined retroperitoneal fatty mass extending through the pelvic outlet into the proximal thigh, displacing adjacent soft tissues without evidence of osseous destruction or vascular invasion.

**Figure 3 reports-08-00181-f003:**
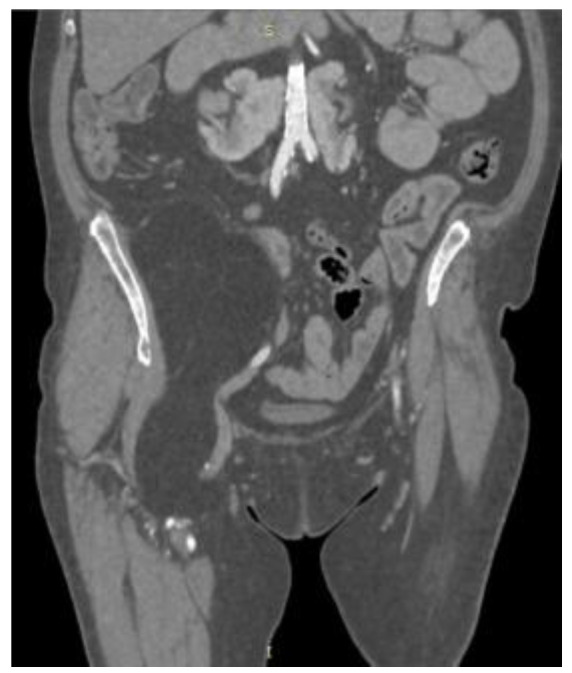
Coronal contrast-enhanced CT scan of the abdomen and pelvis demonstrating a large, well-defined retroperitoneal mass of homogeneous fatty density, displacing adjacent structures without evidence of invasion.

**Figure 4 reports-08-00181-f004:**
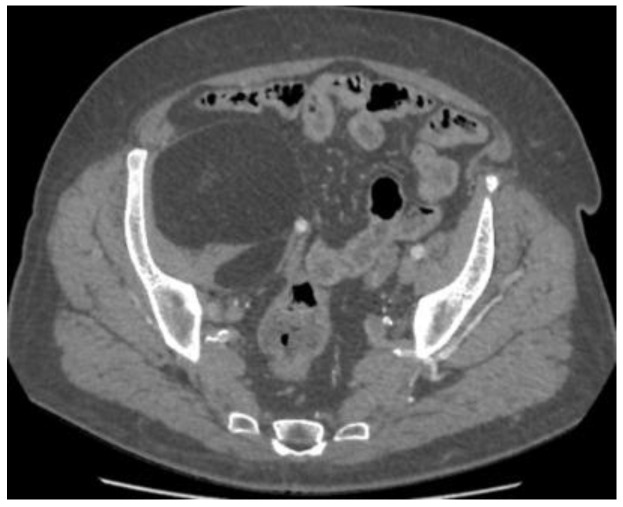
Axial CT section at the pelvic level showing the same lesion, with homogeneous attenuation consistent with adipose tissue and without septations or nodular components suggestive of liposarcoma.

**Figure 5 reports-08-00181-f005:**
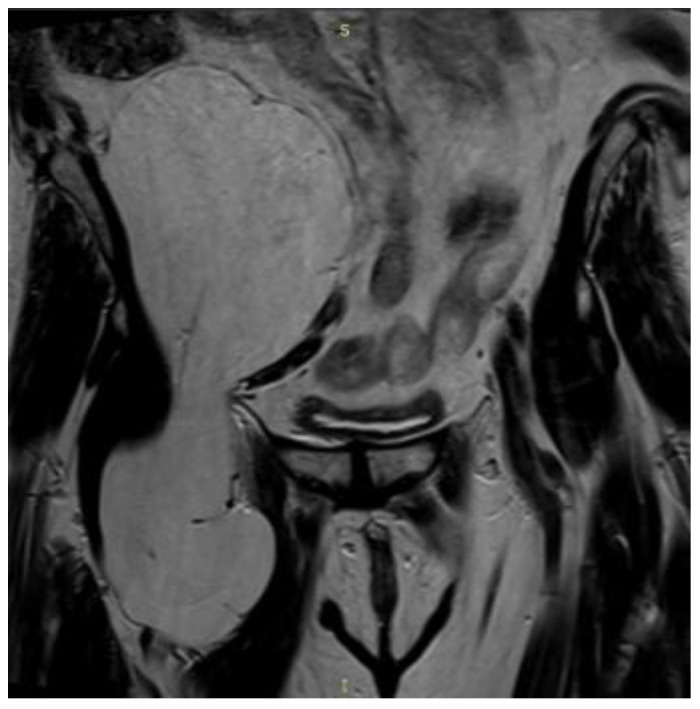
Coronal MRI (T1-weighted sequence) revealing a well-encapsulated lesion of high signal intensity, isointense with subcutaneous fat, confirming the lipomatous nature of the mass.

**Figure 6 reports-08-00181-f006:**
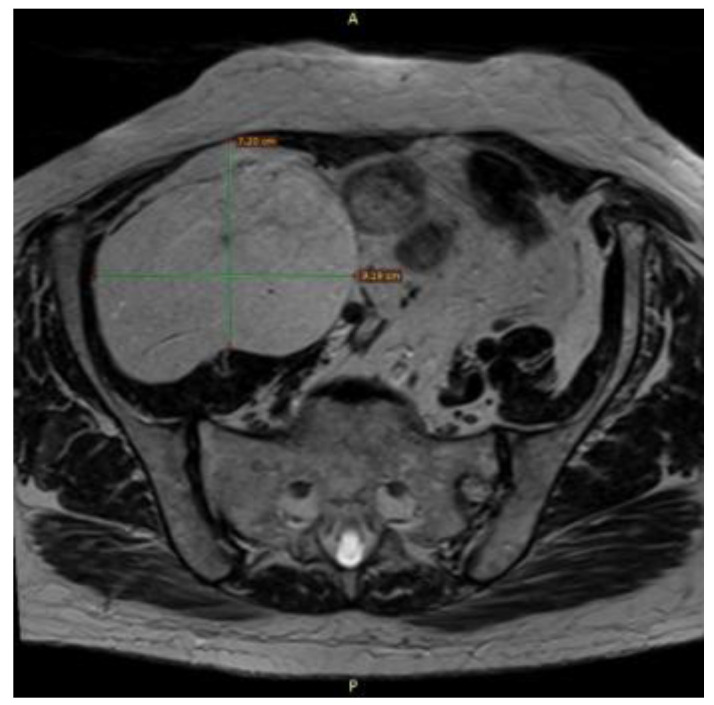
Axial MRI (T1-weighted sequence) showing the retroperitoneal mass 7.2 × 9.19 cm with clear margins and no infiltration of surrounding muscles or viscera.

**Figure 7 reports-08-00181-f007:**
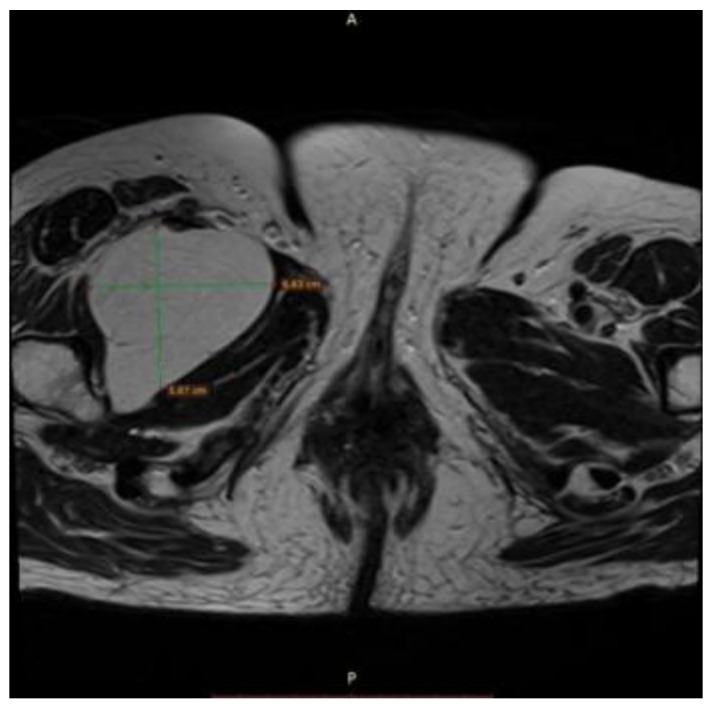
Axial T1-weighted MRI showing a well-circumscribed, homogeneous, hyperintense lipomatous mass in the left pelvic region, measuring approximately 6.6 × 5.9 cm, consistent with adipose tissue signal intensity and without evidence of infiltration into adjacent musculature or viscera.

**Figure 8 reports-08-00181-f008:**
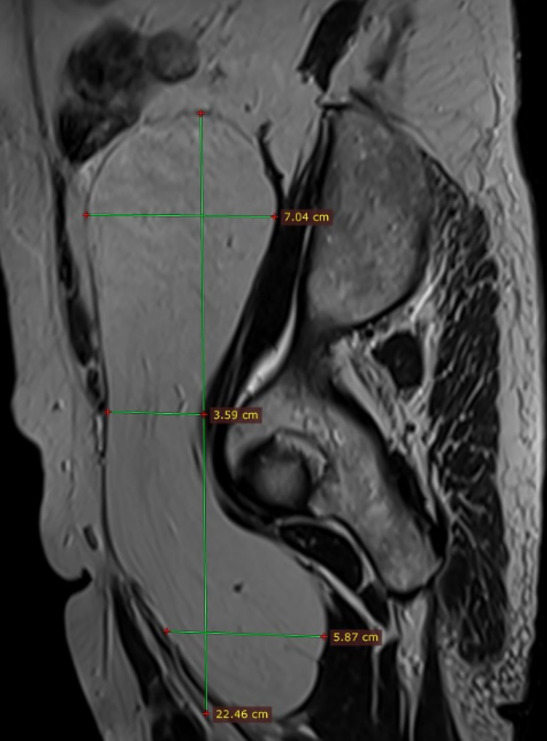
Sagittal T1-weighted MRI showing hyperintense lipomatous mass in the retroperitoneal fat extended throughout the pelvic outlet in the proximal thigh measuring approximately 22.4 cm in length.

**Figure 9 reports-08-00181-f009:**
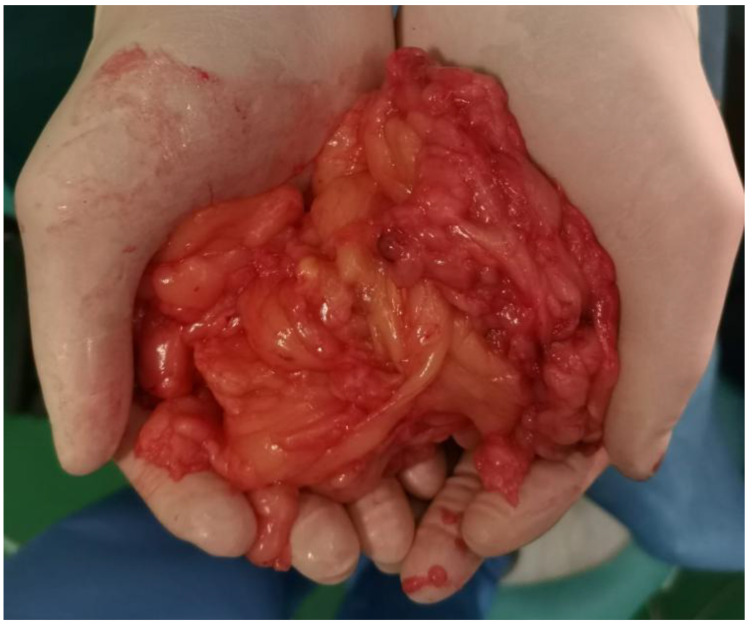
Intraoperative specimen of the excised retroperitoneal mass, displaying a lobulated, yellowish, encapsulated appearance, consistent with a giant lipoma.

## Data Availability

The original contributions presented in this study are included in the article. Further inquiries can be directed to the corresponding author.

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
