# Peer review of "Asymptomatic Retroperitoneal Lipoma with Extension to the Right Anteromedial Thigh"

_reports, 2025, doi:10.3390/reports8030181_

Round 1

Reviewer 1 Report

Comments and Suggestions for Authors

Major Strengths:

This is a well referenced, well documented, single center ,case study which highlights the successful surgical treatment of a rare,retroperitoneal lipoma (approximately 20 cases reported in the literature).

Major Weaknesses:

  1. This is a retrospective case study, so this limits its broad applicability.
  2. The limitations of the case study are not well described.

Specific issues that need to be addressed by author(s):

1,2.I respectfully recommend that the authors add a Limitations section, and briefly discuss the issues associated with a case presentation (Nissen T, Wynn R. The clinical case report: a review of its merits and limitations. BMC Res Notes. 2014 Apr 23;7:264. doi: 10.1186/1756-0500-7-264. PMID: 24758689; PMCID: PMC4001358).

I believe this manuscript adds meaningful clinical information the extant literature.

Once the above edits have been completed, I would  have no further concerns.

I congratulate the authors on their clinical outcome and this, otherwise  excellent case presentation.

I recommend minor revision.

Author Response

Thank you for the review, i added Limitation section and discussion on line 244-250.

Is there anything i should change in the manuscript?

Reviewer 2 Report

Comments and Suggestions for Authors

In this manuscript, Balta et al. address the topic of “Asymptomatic Retroperitoneal Lipoma with Extension to the 2 Right Anteromedial Thigh”. This study is both novel and informative. I have some specific comments for the authors to further improve the case report and manuscript.

  1. Which are the histological subtypes of retroperitoneal tumors exist, and how do they affect prognosis and treatment outcomes?
  2. How do the size and location of retroperitoneal tumors influence the surgical plan and the chances of removing them completely with clear margins?
  3. How do imaging techniques like CT scans and MRI help in accurately diagnosing and determining how advanced retroperitoneal tumors are?
  4. How well do pre-surgery treatments such as radiation and chemo work in helping patients with retroperitoneal sarcomas have better surgical results?
  5. What are the main reasons these tumors come back after treatment, and what surgical or follow-up strategies can help reduce this risk?
  6. What genetic and molecular markers are linked to retroperitoneal sarcomas, and how can they help shape personalized treatment plans?
  7. Does the experience level of the surgeons and the number of cases a hospital handles make a difference in patient outcomes when removing retroperitoneal tumors?

Author Response

1) Histologic subtypes and how they affect prognosis & treatment

The most common retroperitoneal sarcoma subtypes are:

  • Well-differentiated liposarcoma (WDLPS) – indolent but prone to local recurrence; death often results from uncontrolled local disease rather than metastasis. Complete, margin-negative resection is essential.

  • Dedifferentiated liposarcoma (DDLPS) – higher grade, with worse local control and metastatic risk compared to WDLPS; systemic therapy may be considered in advanced cases.

  • Leiomyosarcoma (LMS) – has a greater tendency to metastasize to lungs and liver; systemic therapy is more frequently considered, though surgery remains the main treatment for localized disease.

  • Undifferentiated pleomorphic sarcoma (UPS) and other less common tumors (such as solitary fibrous tumor, malignant peripheral nerve sheath tumor, and myxofibrosarcoma) – prognosis varies, but tumor grade and the completeness of resection are the main outcome determinants.

Histology guides the expected relapse pattern (local versus distant) and informs whether radiotherapy or chemotherapy should be considered.

2) How size and location influence the surgical plan and R0 rates

Large tumors and those that cross into multiple compartments displace or encase adjacent organs, making complete resection difficult. Achieving negative margins is particularly challenging when tumors are close to major vessels, ureters, or bowel.
Location dictates the surgical approach, sometimes requiring combined or staged incisions. Tumors extending through anatomical windows, as in the case of thigh extension, may need dual approaches to protect nerves and vessels.
Positive margins are strongly linked with local recurrence, and the likelihood of achieving clear margins is higher in experienced, high-volume centers.

3) Role of CT and MRI in diagnosis and staging

Contrast-enhanced CT is the standard for defining the tumor’s origin, size, extent, and relation to nearby structures. CT of the chest is used to assess lung metastases.
MRI offers better soft-tissue characterization, defines involvement of fascia and nerves, and is helpful in surgical and radiotherapy planning.
Image-guided biopsy is often performed when sarcoma is suspected to secure a histological diagnosis before considering neoadjuvant treatments.

4) Effect of preoperative treatments

Preoperative radiotherapy has not shown an overall survival benefit for all retroperitoneal sarcomas but may improve local control in selected subtypes such as liposarcoma.
Chemotherapy has a limited role in localized disease and is more often used for high-grade tumors or in metastatic settings. Standard drugs include doxorubicin-based regimens, with newer agents such as eribulin, trabectedin, and pazopanib available for advanced disease.

5) Reasons for recurrence and risk reduction

Recurrence is driven mainly by positive or close surgical margins, large tumor size, multifocal growth, and histology.
Risk can be reduced through surgery in specialized sarcoma centers, where en-bloc resections with or without adjacent organs are more feasible. Preoperative radiotherapy may reduce local recurrence risk in selected patients.
Close postoperative surveillance with scheduled imaging is essential to detect recurrence early and allow reoperation when feasible.

6) Genetic and molecular markers

Well- and dedifferentiated liposarcomas are characterized by MDM2 and CDK4 amplification, which also serve as diagnostic markers. These alterations are being explored as therapeutic targets.
Myxoid liposarcomas harbor a characteristic FUS–DDIT3 fusion, which is associated with radiosensitivity in some settings.
Leiomyosarcomas and undifferentiated pleomorphic sarcomas often show complex genomic alterations with fewer actionable targets, but molecular profiling may guide clinical trial enrollment or personalized therapies.

7) Influence of surgeon and hospital experience

Outcomes are significantly better at high-volume sarcoma centers. These hospitals and surgeons achieve higher rates of complete resections, lower perioperative mortality, and improved long-term survival compared to low-volume centers.

Reviewer 3 Report

Comments and Suggestions for Authors

The authors presented a rare case of retroperitoneal lipoma exteding into the thigh.

Here are a few suggestions which might improve the readability of the manuscript

  1. In the Introduction, it may be helpful to highlight what's not known in this field. Propose a question your manuscript is trying to answer, i.e. the purpose of the study.
  2. In the case presentation, the authors may want to explain why a frozen section was not carried out to rule out liposarcoma.
  3. In the discussion, the authors may consider reviewing the literature, summarising the statistics and suggesting a guideline in the management of similar cases in the future. 
  4. The conclusion should consist a take home message for the reader. 
Comments on the Quality of English Language

Editing by a native English speaker would improve the readability of the manuscript.

Author Response

Hello , here are the answers for your questions.
Is everything ok now?
1) However, there are still knowledge gaps regarding the optimal management of large benign retroperitoneal tumors that mimic malignant behavior. In particular, the extension of retroperitoneal lipomas into adjacent compartments, such as the thigh, raises questions about diagnosis, surgical strategy, and recurrence risk. This case report aims to address these gaps by presenting a rare instance of a giant retroperitoneal lipoma with thigh extension and discussing its clinical and surgical implications.

2) A frozen section was not performed intraoperatively. This decision was based on the characteristic imaging findings (homogeneous fatty density on CT and MRI without suspicious features), the well-encapsulated macroscopic appearance, and the absence of intraoperative signs suggesting sarcomatous transformation. Given these factors, the surgical team proceeded with complete excision and capsule preservation, reserving definitive histopathological examination to confirm the benign nature of the tumor and exclude liposarcoma.

3)
A review of the literature shows that retroperitoneal lipomas account for fewer than 20 published cases, with most patients presenting late due to vague or absent symptoms [5–7,9,11]. Reported tumor sizes vary widely, from 10 cm up to more than 30 cm, and surgical excision remains the cornerstone of treatment. Although most lipomas behave in a benign fashion, the diagnostic challenge lies in differentiating them from well-differentiated liposarcomas, which share overlapping radiological features. Frozen section is often inconclusive in this context, and definitive histopathology remains mandatory [2–4].

Based on available data, a pragmatic approach to similar cases would include: (1) thorough preoperative imaging with CT and MRI to characterize the mass; (2) consideration of biopsy only if sarcoma-directed neoadjuvant therapy is planned, in line with current sarcoma management guidelines; (3) complete surgical excision with preservation of neurovascular structures whenever possible; and (4) structured follow-up with cross-sectional imaging at 6–12 months to monitor for recurrence. This framework can serve as a guideline for clinicians managing rare benign retroperitoneal tumors with atypical extensions, as seen in our case.

4) Take-home message: Early recognition, thorough preoperative planning, and complete excision with vigilant follow-up are key to optimal outcomes in patients with giant retroperitoneal lipomas.

Reviewer 4 Report

Comments and Suggestions for Authors

There are several comments on this manuscript.

1) The introduction should be shortened to a large extent because the extended information presented here belongs to the discussion section of the manuscript (histology, presentation, diagnostic approach, treatment options and prognosis should be removed to the discussion).

2) The presentation of the case is poor and needs to be re-written. More specifically:

Why the patient did not have treatment 2 years earlier when the lipomatous tumor of the right thigh 84 was diagnosed by CT?

Any details about other lipomas or lipomatosis in this patient?

Trombex 75 mg: please give the active substance instead of the commercial name, and cardiological treatment: which one?

CT and MRI findings should be presented along with the possible diagnosis and differential diagnosis mainly regarding liposarcoma.

Figure numbers are not mentioned in the text.

There are no figure legends.

What do you mean by “Monoplanar laparography”?

Figures 10 and 11 are superfluous and should be omitted.

Legend of Figure 12: It is not clear to what “Journal of Surgical Case Reports” refers. Is it taken from that Journal? Did the authors got permission?

In the Abstract the authors state that “The patient’s postoperative course was favorable, with minimal residual paresthesia and complete wound healing.”

However, this information is not available in the presentation of the case. It should be included there and be more detailed.

How long was the postoperative follow-up?

Did the resected the tumor by two different incisions? Why they did employ the Karakousis's abdomino-inguinal incision?

3) A proper discussion is lacking. Existed information on presentation, diagnostic approach, treatment options, histology and prognosis should be included the discussion. To this extend, preoperative imaging, probable biopsy results and intra-operative evaluation of tumor’s anatomical features are critical for correct surgical planning especially if malignancy cannot be ruled out. The differential diagnosis mainly with well-differentiated liposarcoma which is the most frequent histological type in the retroperitoneum along with dedifferentiated liposarcoma is of upmost importance.

It should be also highlighted that complete surgical resection is the mainstay treatment and a close and periodic follow-up is necessary.

4) The reference list should be updated and to include s similar cases presented in the literature. See:

Fernández Hernández JÁ, et al. Giant lipomas or retroperitoneal liposarcomas? Controversies in their diagnosis and treatment. Rev Esp Patol. 2021 Apr-Jun;54(2):75-84.

Nardi WS, et al. Resection of a giant retroperitoneal lipoma herniating through the inguinal canal.  BMJ Case Rep. 2021 Jan 26;14(1):e239301.

Selmani R, et al. Giant retroperitoneal liposarcoma: а case report. Prilozi. 2011;32(1):323-32.

Petca RC, et al. Half abdomen tumor - giant retroperitoneal lipoma: a case report and review of the literature. Rom J Morphol Embryol. 2022 Jan-Mar;63(1):237-244.

Kotohata Y, et al. Giant retroperitoneal lipoma resulting in femoral hernia: A case report Int J Surg Case Rep. 2025 Aug;133:111584.

Chen ZY, et al. Giant retroperitoneal lipoma presenting with abdominal distention: A case report and review of the literature. World J Clin Cases. 2022 Feb 16;10(5):1675-1683.

Author Response

1. Why the patient did not have treatment 2 years earlier?

Because the patient refused surgery

2. Any details about other lipomas or lipomatosis?

 The patient had no other lipomas or signs of systemic lipomatosis.

3. Trombex 75 mg: active substance and cardiological treatment

I changed Trombex with Clopidrogrel and antihypertesion medication

4. CT and MRI findings, possible diagnosis, differential diagnosis

DD was not considered due to images from CT and MRI confirming the lipomatous structure from lipoma (beningn tumour) with no invasion in other structures

5. Figure numbers and legends

I deleted the fig 10 and 11 , and added description to legends

6. “Monoplanar laparography”

Monoplanar laparoraphy- changed it in manuscript 

7. Postoperative course details and follow-up

Postoperative follow up was 1 month postoperatively, needing to to a CT scan/MRI in 6 months

8. Two different incisions and Karakousis abdomino-inguinal incision

Incisions were made due to imposibility to extract the lipomatous tumour through 1 incision

Discussions : 

Retroperitoneal tumors are a rare pathology that requires careful evaluation at diagnosis, with timely surgical intervention, ideally preceded by biopsy. Their progression is often asymptomatic, allowing them to reach a considerable size. The main reasons patients seek medical care are compression of adjacent organs or mechanoreceptors.

Failure to perform surgery promptly can complicate the approach and prevent complete excision. In this case, the tumor was resected via two approaches, with capsular breach, which increases the risk of local recurrence. Therefore, imaging follow-up at 6–12 months postoperatively is necessary. Our case is quite similar to the case published by Jason R Laurens et al [11].

A review of the literature shows that retroperitoneal lipomas account for fewer than 20 published cases, with most patients presenting late due to vague or absent symptoms [5–7,9,11]. Reported tumor sizes vary widely, from 10 cm up to more than 30 cm, and surgical excision remains the cornerstone of treatment. Although most lipomas behave in a benign fashion, the diagnostic challenge lies in differentiating them from well-differentiated liposarcomas, which share overlapping radiological features. Frozen section is often inconclusive in this context, and definitive histopathology remains mandatory [2–4].

Based on available data, a pragmatic approach to similar cases would include: (1) thorough preoperative imaging with CT and MRI to characterize the mass; (2) consideration of biopsy only if sarcoma-directed neoadjuvant therapy is planned, in line with current sarcoma management guidelines; (3) complete surgical excision with preservation of neurovascular structures whenever possible; and (4) structured follow-up with cross-sectional imaging at 6–12 months to monitor for recurrence. This framework can serve as a guideline for clinicians managing rare benign retroperitoneal tumors with atypical extensions, as seen in our case.

I added references. 

Thank you, any other changes should i make?

Reviewer 5 Report

Comments and Suggestions for Authors

The authors present a rare and clinically valuable case of an asymptomatic retroperitoneal lipoma extending into the right anteromedial thigh. The case is well described and the manuscript is well organized. The dual surgical approach is appropriately justified given the tumor’s anatomical extent, and the favorable postoperative course further highlights the importance of multidisciplinary planning. The inclusion of imaging findings, intraoperative steps, and histological features contributes to the clinical relevance of this report, particularly for surgeons dealing with retroperitoneal pathology that mimics inguinal or femoral hernia.

Nevertheless, the manuscript would benefit from some refinements. First, the introduction could be strengthened by briefly discussing differential diagnoses in similar clinical presentations. Additionally, while the dual approach is described clearly, a brief schematic or intraoperative illustration (if available) might enhance clarity. Minor grammatical improvements in the discussion and abstract could improve overall readability. Lastly, given the rarity of this anatomical extension, a more thorough comparison with previously published cases—including Laurens et al.—could further highlight the uniqueness and clinical contribution of this report. Overall, this is a valuable contribution that merits publication with minor revisions.

Author Response

Hello, i made some changes in the Manuscript. If there is other what you recommend to change, feel free to guide me. Thanks

Round 2

Reviewer 3 Report

Comments and Suggestions for Authors

The authors have made appropriate amendments

Author Response

hello, i uploaded the modified version. thanks

Reviewer 4 Report

Comments and Suggestions for Authors

In the revised manuscript have made substantial changes according to the reviewer’s comments and there is considerable improvement.

However, there some further points that need attention in the revised manuscript:

The introduction is very long and should be shortened giving only essential information as an introduction to the subject. Many of the information presented here belong to the discussion section. Actions should be taken accordingly.

The order of appearance of references in the manuscript seems to not follow the order of their first appearance. Attention should be given in their properly mentioning and correct numbering. To mention just some examples: Ref. 13-18 shown in the introduction are not mentioned in the discussion. Ref. 11 is not shown in the introduction but in the discussion.

The figures are not mentioned in the presentation of the case.

The figure legends should be more detailed and descriptive.

The figures should be numbered correctly. See Fig 6. which is mentioned twice. There is no figure numbered as 8.

Figure 10. is taken from the paper of Laurens JR et al. The legend is not appropriate. It does not add anything to this case report and should be omitted especially if no permission by the author was obtained.

Author Response

Hello, I have uploaded the response to the comments file. Thank you.

Round 3

Reviewer 4 Report

Comments and Suggestions for Authors

The authors responded satisfactory to the comments raised by the reviewer.

The manuscript is now better organized, focused and in accordance with case report presentations.

There are some minor points to be addressed.

1) Language editing would be useful especially regarding the use of tenses.

2) The presentation of the references in the reference list should be uniform and must follow the established rules.

Author Response

1) Language editing would be useful especially regarding the use of tenses.

Answer: Thank you for your observation. Based on the first two reports, no issues with tense usage were identified. Therefore, it does not appear to be a consistent problem. Nonetheless, we will remain attentive to this aspect to ensure clarity and accuracy in language.

2) The presentation of the references in the reference list should be uniform and must follow the established rules.

Answer: I reorganized the reference list, thank you
